# Dental Loupe’s Role in Detection of Caries of Molars in Children by Student and Dentists

**DOI:** 10.3390/children10061050

**Published:** 2023-06-12

**Authors:** Sigalit Blumer, Siwar Zidani, Benjamin Peretz, Lazar Kats, Hanaa Azem, Sarit Naishlos, Johnny Kharouba

**Affiliations:** 1Department of Paediatric Dentistry, The Maurice and Gabriela Goldschleger School of Dental Medicine, Faculty of Medicine, Tel Aviv University, Tel Aviv 69978, Israel; 2Department of Oral Pathology, Oral Medicine and Maxillofacial Imaging, The Maurice and Gabriela Goldschleger School of Dental Medicine, Faculty of Medicine, Tel Aviv University, Tel Aviv 69978, Israel

**Keywords:** dental loupes, student caries diagnosis, caries diagnosis primary teeth, visual examination, caries detection

## Abstract

Purpose: The objectives of this study were to examine the effectiveness of diagnosing occlusal caries in molar teeth in children under the use of loupes and, secondarily, to examine whether there is a difference in the diagnosis between permanent and primary teeth using dental loupes. In addition, to check whether the student’s diagnosis using loupes improves caries diagnosis compared to dentists’ diagnosis in both methods. Methods: The data were collected from 163 patients aged 6–14 who sought treatment in the Pedodontic Department of the Faculty of Dentistry at Tel-Aviv University during 2020–2021. The first and second permanent molars and second primary molars with deep groves were examined. A student and dentists made the diagnosis with and without loupes while using the ICDAS criteria. Results: The student’s examinations without the loupes detected no caries in 60% of the cases compared to 76.9% in the examinations with the loupes and found initial caries without cavitation (ICDAS1) in only 17.6% of teeth without loupes examination compared to 33% using loupes. The dentist correctly diagnosed no caries (ICDAS0) in 82.1% of cases without loupes and initial caries without cavitation (ICDAS1) in 62.5% of cases. The dentist correctly diagnosed distinct caries without cavitation (ICDAS2) in 90.8% of cases. No differences were observed in caries diagnosis between primary and permanent teeth when the examiner was a specialist/intern using loupes; however, there was a statistically significant difference (*p* = 0.047) when the diagnosis was made by a student using loupes. Clinical significance: The use of dental loupes is an effective method for the correct and early diagnosis of occlusal caries lesions in children’s molar teeth by both dentists and students, and this is in accordance with the principle of minimally invasive dentistry. There is a justification for the use of dental loupes for the diagnosis of initial occlusal caries in primary and permanent molars in children in a precise way. Using loupes especially improves the correct diagnosis of initial caries in primary teeth by students.

## 1. Introduction

Dental caries is a complex oral condition influenced by biofilm formation, dietary factors, and various contributing factors. It is a non-communicable disease characterized by the progressive demineralization of dental hard tissues. The clinical manifestation of dental caries is observed as caries lesions, which can be distinguished based on their anatomical position (coronal or root/cementum surface), the extent of damage (non-cavitated or cavitated), depth of tissue involvement (enamel, dentin, pulp), and their level of activity (active or inactive).

Early detection, assessment, and diagnosis of caries lesions are essential and allow dentists to move from invasive to non-invasive conservative treatment [1]. The greatest challenge in treating caries is controlling the progression of the disease using non-invasive procedures and early diagnosis [2]. In children and adolescents, tooth decay mainly occurs on occlusal surfaces [3]. Diagnosis of occlusal caries is a clinical challenge due to the morphological complexity of the groove and fissure system and the frequent presence of pigmentation in the grooves [4]. Non-invasive methods for diagnosing and treating tooth decay and preserving dental tissue have recently emerged as an alternative to standard invasive treatment [5].

Many diagnostic techniques for detecting caries have been developed, such as Quantitative Light Fluorescence (QLF), Electrical Conductivity Measurements (ECM), and Digital Imaging Fiber-Optic Trans-Illumination (DFOTI), among others. Their role, in general, is to detect more and smaller lesions, but visual inspection remains the most widely used technique [6]. Detecting tooth decay by visual examination includes identifying areas that have undergone demineralization, namely white spot lesions (WSL) on pits and grooves suspected to be infected with caries [7].

Numerous factors can prevent early diagnosis of caries on visual inspection, such as plaque, inappropriate light source, difficult access, enamel defects, and the subjectivity of the method [7]. Lack of cooperation during visual inspection of young children with primary teeth can also prevent accurate diagnosis. Visual detection of occlusal caries has low sensitivity and high specificity. The authors of a comprehensive review estimated visual diagnosis to have a sensitivity of 39–59% in both enamel and dentine of occlusal surfaces. Specificity, however, was around 95% [8]. A scoring system, such as the International Caries Detection and Assessment System (ICDAS), has been shown to be more accurate than traditional methods, such as the World Health Organization (WHO) system [9], and increases the likelihood of identifying occlusal dentin lesions [by up to 40%] from below 50% to between 55% and 90% [9]. The ICDAS method is based on a study by Ekstrand et al. in which the authors developed a rigorous, accurate visual description that determined the depth of the caries lesion, which was then verified by a histological examination [10].

One study [11] tested the effect of different magnifications in the diagnosis of dental caries. Permanent molars were examined with the naked eye and at magnifications of 2.5×, 4.5×, 10×.

In another study [12], posterior teeth were tested under laboratory conditions. The results of this study showed that the use of loupes with ×1.8 magnification for caries diagnosis did not increase, and even decreased, the sensitivity.

Another study [1] compared the effectiveness of visual examination and examination under low magnification in the detection of dental caries in a dental outreach program. It concluded that the difference in the effectiveness of vision under low magnification and normal visual examination in detecting dental caries in a dental outreach program is not significant.

The utilization of dental loupes has demonstrated the potential to enhance the precision of visual inspections, particularly in cases involving primary teeth or when conducted by dental students or less experienced dentists. The application of loupes in general and paediatric dentistry has witnessed a growing trend, yet research investigating their impact on diagnosing tooth decay in primary and permanent teeth of paediatric patients remains absent.

The primary endpoints of this study were to assess the effectiveness of diagnosing dental caries in children using ×2.5 loupes without lighting compared to a visual examination without loupes and to determine whether the diagnosis of tooth decay differs between permanent and primary teeth using loupes.

## 2. Materials and Methods

### 2.1. Study Design

To analyse the study endpoints, the diagnosis of occlusal tooth decay made without loupes was compared to the same diagnosis made for the same tooth under a ×2.5 dental loupe without lighting. The diagnosis was performed by a dentistry student and two dentists (intern and specialist) at the Paediatric Dental Department of the Tel Aviv University School of Dentistry. The study obtained the approval of the ethics committee of the Tel Aviv University School of Dentistry, Institutional Review Board protocol number No. 0000373-3, issued on 2 April 2020.

### 2.2. Sample Size

G-power software was used to calculate the sample size. Assumptions for calculation: Type 1 (alpha) error of 0.05, minimum test intensity 80%. An additional assumption for the purpose of calculation was that the magnitude of the expected effect of the difference in caries diagnosis between visual detection without loupes and diagnosis with dental loupes would be at least 0.2 [11]. Based on these assumptions, the minimum sample size was 156 teeth.

### 2.3. Patient Population

One hundred sixty-three patients were recruited between November 2020 to May 2021 for this study. The participants, aged 6–14, were treated in the paediatric dental department of the Tel Aviv University School of Dentistry. The mean age was 9.46 years (SD 2.33). Signed consent forms by their parents were obtained. Most patients (60.1%) had first and second permanent molars, and the rest (39.9%) had second primary molars.

The diagnoses were performed by a team consisting of a dental student, a specialist in paediatric dentistry, and an intern in paediatric dentistry. Each patient was diagnosed by the paediatric dentist, with and without loupes, by the intern with and without loupes, and by the student with and without loupes. To obtain more reliable test results, the diagnoses obtained by the specialist (paediatric dentist) and the intern were calibrated and grouped into a single item, called the dentist, for the purpose of analysis.

The inclusion criteria were permanent and primary molars with an intact occlusal surface, with caries lesions without cavitation, and molars with deep fissures.

The exclusion criteria were teeth with occlusal surfaces that include restorations or sealants, teeth that show the collapse of the enamel in the occlusal surfaces, teeth with developmental or hereditary defects such as Molar incisor hypomineralisation (MIH) and Amelogenesis imperfecta.

The gold standard was the diagnosis made by the paediatric dentist with loupes. The molars were cleaned with prophylaxis paste. Visual inspection was first performed without drying the tooth and then again after drying the tooth surface for 5 s. The visual examination was performed without probing, with the patients sitting in a chair under normal clinical lighting. Water/air spray and a flat dental mirror were used. The tests were performed by the student, the paediatric dentist, and the intern during the same session.

To reduce bias, 50% of the tests were performed first without loupes and then with ×2.5 loupes without lighting, and the other half were performed first with ×2.5 loupes without lighting and then without loupes. The criteria used to describe the findings in the visual examination were based on the ICDAS grading method and are shown in Table 1 [8].

### 2.4. Data Analysis

All data were entered into SPSS version 25 for statistical analysis. Categorical variables (e.g., sex) were analysed using descriptive statistics (frequency, percentage), and continuous variables (e.g., age) were analysed using the range, mean, and standard deviation. Next, continuous variables were tested for normal distribution using the Shapiro-Wilk test, and based on the findings of these preliminary tests, follow-up tests were used to determine the primary endpoints.

The reliability of diagnosing caries without loupes vs. under loupes was tested using the Kappa index. The higher this index, the more uniform and consistent the diagnoses made by the various scales. The kappa value was calculated as 0.78. 

Differences between the diagnosis of caries by visual examination without the use of loupes and the diagnosis of caries by loupes were examined using McNemar’s test. McNemar’s test calculates differences in the diagnosis frequency for the same tooth in two different situations. The minimum significance level was set at *p* < 0.05. Differences between groups were assessed using Chi-square tests and Fisher Exact tests.

## 3. Results

Table 2 shows the diagnosis made by the student and dentist (intern and specialist) according to the use of loupes. Table 2 demonstrates that the student diagnosed with loupes 10.4% of teeth as ICDAS1 (initial caries without cavitation), 68.1% as ICDAS2 (distinct caries without cavitation), and the rest (21.5%) as ICDAS0 (no caries). Without loupes, however, the student diagnosed 7.4% of teeth as ICDAS1 (initial caries without cavitation), 73.6% as ICDAS2 (distinct caries without cavitation), and 16.0% as ICDAS0 (no caries). The student with loupes also diagnosed five cases (3.1%) as ICDAS3 (caries with cavitation located only in the enamel). These five cases needed restorations.

The dentist diagnosed 9.8% of teeth as ICDAS1, 73.0% as ICDAS2, and 17.2% as ICDAS0 using loupes. Without loupes, the dentist diagnosed 9.2% as ICDAS1, 69.9% as ICDAS2, and 14.1% as ICDAS0. The dentist also diagnosed 11 (6.7%) cases with cavitation located only in the enamel (ICDAS3).

Table 2 also shows the effectiveness of diagnosing occlusal caries in children. The student correctly diagnosed no caries (ICDAS0) in 60% of cases without loupes compared to 76.9% of cases using loupes and found initial caries without cavitation (ICDAS1) in only 17.6% of teeth without loupes compared to 33% using loupes. The dentist (without loupes) diagnosed 62.5% as ICDAS1 and 82.1% as ICDAS0. The differences in the above results were found to be statistically significant (*p* < 0.01). In the diagnosis of distinct caries without cavitation (ICDAS2), the student correctly diagnosed 87.4% of cases without loupes and 90% with loupes. The difference was not statistically significant.

The dentist correctly diagnosed no caries (ICDAS0) in 82.1% of cases without loupes and initial caries without cavitation (ICDAS1) in 62.5% of cases. The dentist correctly diagnosed distinct caries without cavitation (ICDAS2) in 90.8% of cases.

The above results lead us to conclude that the diagnosis of caries with loupes is more effective and accurate compared to a visual diagnosis without loupes for both the student and the dentist and that there is a statistically significant relationship between the use of loupes and the correct diagnosis of caries (*p* < 0.01), for both the student and the dentist.

### Difference between Diagnosis of Primary and Permanent Teeth Using Loupes

Chi-square tests were performed to distinguish differences in diagnosis between primary and permanent teeth using loupes. Data were collected from the student and dentist separately. Table 3 shows the distribution of all diagnoses by primary and permanent teeth.

As shown in Table 4, there were no differences in diagnosis between primary and permanent teeth when made by the dentist using loupes (χ^2^ (3) = 3.12. *p* = 0.37). However, there was a significant difference when made by a student using loupes (χ^2^ (3) = 7.97. *p* < 0.05), in that as a higher percentage of permanent teeth (11.2%) were diagnosed with initial caries without cavitation (ICDAS1) compared to primary teeth (1.5%).

Table 4 compares primary and permanent teeth diagnoses of ICDAS1, ICDAS2, and ICDAS 1 + 2 performed by students and dentists using loupes. Results show that no differences were found between students and dentists in most of the comparisons. However, for ICDAS1, higher rates of detection were found for dentists compared with students (*p* < 0.001).

## 4. Discussion

The results of this study display that the diagnosis of caries with loupes was more effective and accurate compared to visual examination without loupes by both the student and the dentist (intern and specialist). In addition, a statistically significant relationship was found between the use of loupes and early diagnosis of caries lesion or absence of caries (*p* < 0.01). According to the results, the dentist’s diagnosis was superior to the student’s diagnosis both with and without loupes, and examination under loupes improved the diagnosis of tooth decay by the student. There were no differences between primary and permanent teeth when the diagnosis was made by the dentist using loupes, but there was a significant difference when the diagnosis was made by a student using loupes.

The goal of using advanced techniques for the diagnosis of caries is to improve the sensitivity and specificity of the conventional visual test. In other words, the aim is to identify a greater number of carious lesions in their early stages while ensuring that the test’s ability to accurately exclude the presence of caries lesions is not compromised. The findings of the study indicated improved diagnostic capabilities for early carious lesions, as well as accurate identification of non-carious lesions. These results hold true to a significant extent, particularly for dental students and in the context of diagnosing caries lesions in primary teeth.

Early diagnosis of caries is the cornerstone of minimally invasive dentistry. The findings of the present study indicate that the use of dental loupes is effective not only for dental students but also for young dentists and specialists. In paediatric dentistry, dental procedures often necessitate behaviour management techniques, including the use of sedation or general anaesthesia. Consequently, early detection of caries through the application of dental loupes in children can help prevent unnecessary invasive interventions. Instead, conservative treatments such as fluoride application or fissure sealants can be implemented, supported by behaviour management techniques, such as “tell, show, and do” to facilitate the process. This approach promotes minimal intervention and preserves the dental health of children effectively.

Early diagnosis of caries lesion without cavitation stages of ICDAS1 and ICDAS2 enables conservative treatment, such as follow-up and applying a fluoride gel every three months. The early diagnosis especially helps students who have little experience and when it comes to the diagnosis of caries in primary teeth in young children who often do not cooperate during a dental examination.

A study by Neuhaus et al. [11] examined the effect of various magnifications on the diagnosis of dental caries in 100 extracted molars examined by 14 examiners, including dental students and general dentists. As in the current study, the test was based on ICDAS criteria. However, in Neuhaus et al. study, the gold standard was a histological test, which is more accurate than the standard used in the present study. The extracted teeth were examined macroscopically and by using three different magnifications: ×2.5, ×4.5, and ×10. The results showed that the greater the magnification, the fewer the number of teeth diagnosed without caries and the greater the number of teeth diagnosed with caries with local enamel cavitation (ICDAS3). The authors concluded that the optimal range for detecting caries according to the ICDAS criteria is between 0 and ×2.5 magnification. Beyond that, the likelihood of unnecessary invasive intervention increases. This report is consistent with the results of the present study in that magnification up to ×2.5 is effective in diagnosing dental caries and shows that greater magnification may cause more harm than good and lead to unnecessary invasive interventions.

Sisodia and Manjunath [12] examined 60 posterior teeth (molars and premolars) under laboratory conditions using three different methods: no magnification, ×1.8 magnifications, and under a surgical microscope at ×3.4 magnification. The results showed that using ×1.8 loupes to diagnose tooth decay reduced sensitivity from 0.71 (no magnification) to 0.42 but increased specificity from 0.72 (no magnification) to 0.9.

In another study [13], Gupta et al. evaluated the efficacy and reliability of magnifying loupes and DIAGNOdent in diagnosing White Spot Lesions (WSLs) on smooth surfaces, unlike our study that examined fissures and grooves on occlusal surfaces. In the study in question, 300 children aged 5–10 years were examined for the presence of WSL lesions on smooth surfaces on two occasions by two examiners, once with the naked eye and once with loupes, following which caries were identified using DIAGNOdent. The tests were performed using the ICDAS criteria. The results of that study reported a significant difference between visual examination and visual examination using loupes in both dry and wet conditions (*p* < 0.05), while no significant difference was found between DIAGNOdent and loupe diagnosis. The authors concluded that loupes are effective in diagnosing WSL on smooth surfaces. Despite the differences in primary outcome with the current study, their results confirm the observation of the present study that dental loupes are effective in the early diagnosis of caries lesions.

Goel et al. [14] examined the effect of magnification on the diagnosis of caries lesions in 44 extracted premolar teeth with intact occlusal surfaces and no visible cavitation. Diagnosis was made by visual inspection without magnification (unaided), ×4.2 magnification loupes, and a ×10 magnification microscope. After examination, the teeth were cut buco-lingually, and both surfaces were examined under a ×50 magnification microscope to determine the presence or absence of caries lesions in the fissures and grooves. The loupe examination showed the highest sensitivity and the lowest specificity compared to other methods. This again emphasizes that the use of loupes helps diagnose early caries lesions that might otherwise be missed; however, there is a higher probability of false positive diagnosis and unnecessary intervention. The foregoing study reinforces the results of the current analysis in that the use of loupes is an effective method in the early diagnosis of caries lesions in grooves and thus helps reduce invasive treatments in children. Unlike the present study, Goel et al. used higher magnifications and performed the study in-vitro.

In the current study, no differences were observed in caries diagnosis between primary and permanent teeth when the examiner was a specialist/intern using loupes; however, there was a statistically significant difference (*p* < 0.047) when the diagnosis was made by a student using loupes. Hence, there was a higher percentage of non-cavitated caries diagnosis (ICDAS1) in permanent teeth (11.2%) compared to primary teeth (1.5%). This means that sensitivity increased when the students used loupes. The differences in the student’s diagnosis between primary and permanent teeth appear to be due to inexperience. This is particularly important in young, uncooperative children who are hard to examine. The results suggest that using loupes is effective for all dentists, particularly beginners who have not yet gained much experience or students during their clinical practice. A database search did not reveal similar studies that compared primary and permanent teeth using loupes.

One study [15] showed that the use of loupes is more prevalent among general dentists (64.3%) compared to paediatric specialists and residents (35.7%). Perhaps the appearance of paediatric dentists wearing loupes during treatment might hinder their attempts at effective contact and good rapport, marked by confidence and trust in a relaxed and safe atmosphere. This study also showed that the more a dentist is aware of the benefits of using loupes, the more inclined they will be to use them. The results of the present study emphasize the advantages of using loupes during the child’s dental examination and help in adjusting a preventive and conservative treatment plan.

The current study has certain limitations that should be acknowledged. Firstly, the chosen gold standard for caries diagnosis, which relied on the dentist’s examination using loupes, is subjective in nature and may not be considered ideal. A more reliable approach would have involved histological tests to determine the presence or absence of caries. Secondly, the study did not differentiate between interns and specialists, although it is worth noting that interns typically have a minimum of three years of general experience prior to entering the internship program. The participants in this study were in their final year of the internship program.

The study’s novelty lies in its comparison between primary and permanent teeth, revealing discrepancies in the diagnosis of caries between these tooth types. Additionally, the study highlights differences in diagnostic accuracy when comparing dentists to dental students. These findings contribute valuable insights into the variations in caries diagnosis across different tooth types and between experienced dentists and students.

## 5. Conclusions

The use of loupes has proven to be effective in the early detection of caries lesions, enabling the implementation of preventive dentistry measures and minimizing the need for invasive interventions and unnecessary operative treatments. Magnification of up to 2.5 times has shown effectiveness in diagnosing dental caries, particularly when performed by dental students and in the context of primary teeth. However, further research is warranted to explore the potential of loupes in diagnosing caries, specifically in children. This would provide valuable insights into optimizing diagnostic approaches for caries detection in pediatric dental care.

## Figures and Tables

**Table 1 children-10-01050-t001:** International Caries Detection and Assessment System (ICDAS) [8].

Code	Criterion
0	Sound tooth surface: no evidence of caries after 5 s air drying
1	First visual change in enamel: opacity or discoloration (white or brown) is visible at the entrance to the pit or fissure seen after prolonged air drying
2	Distinct visual change in enamel visible when wet; lesion must be visible when dry
3	Localized enamel breakdown (without clinical visual signs of dentinal involvement) seen when wet and after prolonged drying
4	Underlying dark shadow from dentine
5	Distinct cavity with visible dentine
6	Extensive (more than half the surface) distinct cavity with visible dentine

**Table 2 children-10-01050-t002:** Visual diagnosis and correct diagnosis of occlusal caries in children compared to the gold standard.

Diagnosis	ICDAS0 *n* (%)	ICDAS1 *n* (%)	ICDAS2 *n* (%)	ICDAS3 *n* (%)	ϕ_c_	*p* *	χ^2^ (df)
Student visual diagnosis							
With loupes	35 (21.5%)	17 (10.4%)	111 (68.1%)	5 (3.1%)	0.42	<0.01	89.33 (6)
Without loupes	26 (16.0%)	12 (7.4%)	120 (73.6%)	0	0.51	<0.01	91.42 (6)
Dentist visual diagnosis							
With loupes	28 (17.2%)	16 (9.8%)	119 (73.0%)	0	0.62	<0.01	104.45 (6)
Without loupes	23 (14.1%)	15 (9.2%)	114 (69.9%)	11 (6.7%)	0.67	<0.01	103.58 (6)
Student correct diagnosis							
Without loupes	21 (60%)	3 (17.6%)	97 (87.4%)	0	0.58	<0.01	110.23 (6)
With loupes	20 (76.9%)	4 (33.3%)	108 (90.0%)	5 (100%)	0.64	<0.01	200.09 (6)
Dentist correct diagnosis							
Without loupes	23 (82.1%)	10 (62.5%)	108 (90.8%)	0	0.80	<0.01	206.94 (6)

The gold standard was defined as the diagnosis made by the paediatric dentist with loupes. Data presented as frequency *n* and percentage (%). Dentist stands for the combined results of the paediatric dentist and the intern in paediatric dentistry. ICDAS: International Caries Detection and Assessment System. ϕ_c_ correlation coefficient > 0.5 is considered strong. * *p* value < 0.05 was considered statistically significant.

**Table 3 children-10-01050-t003:** Chi-square tests were performed to distinguish differences in diagnosis between primary and permanent teeth with and without loupes use.

	Permanent Teeth *n* (%)		**Primary Teeth** *n* (%)	
	**Students**	**Dentists**	**Students**	**Dentists**
ICDAS0	12 (12.2%)	12 (12.2%)	14 (21.5%)	11 (16.9%)
ICDAS1	11 (11.2%)	10 (10.2%)	1 (1.5%)	5 (7.7%)
ICDAS2	71 (72.4%)	67 (68.4%)	49 (75.4%)	47 (72.3%)
ICDAS3	4 (4.1%)	9 (9.2%)	1 (1.5%)	2 (3.1%)
# Students	χ^2^ 7.97 *p* 0.47 ϕ_c_ 0.22			
## Dentists	χ^2^ 3.12 *p* 0.37 ϕ_c_ 0.14			

Data presented as frequency *n* and percentage (%). *p* value < 0.05 was considered statistically significant. ϕ_c_ correlation coefficient > 0.5 is considered strong. ICDAS: International Caries Detection and Assessment System. # Differences in diagnoses for the primary or permanent tooth when the diagnostician was a student. ## Differences in diagnoses for the primary or permanent tooth when the diagnostician was a dentist.

**Table 4 children-10-01050-t004:** Comparison between primary and permanent teeth diagnosis of ICDAS1, ICDAS2, and ICDAS1 + 2 performed by students and dentists using loupes.

	Permanent Teeth *n* (%)				Primary Teeth *n* (%)			
	Students	Dentists	χ^2^	*p*	Students	Dentists	χ^2^	*p*
ICDAS1	11 (47.8)%	10 (45.5%)	0.82	0.782	1 (6.7%)	5 (31.3%)	14.23	<0.001
ICDAS2	71 (85.5)%	67 (84.8%)	0.34	0.889	49 (77.8%)	47 (81.0%)	1.03	0.304
ICDAS1 + 2	82 (87.2)%	77 (86.5%)	0.28	0.972	50 (78.1%)	52 (82.5%)	1.21	0.291

ICDAS: International Caries Detection and Assessment System. ICDAS1 + 2 represents caries without cavitation. Data presented as frequency *n* and percentage (%). *p* value < 0.05 was considered statistically significant. ϕ_c_ correlation coefficient > 0.5 is considered strong.

## Data Availability

The data presented in this study are available on request from the corresponding author. The data are not publicly available due to confidentiality issues.

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
