# Peer review of "Dental Loupe’s Role in Detection of Caries of Molars in Children by Student and Dentists"

_children, 2023, doi:10.3390/children10061050_

Round 1
Reviewer 1 Report
Comments to the Authors
This study aimed to examine the effectiveness of diagnosing occlusal caries in molar teeth in children under the use of loupes by student and dentist. However, there are some mistakes in the article.
The following issues need to be improved and clarified:
1. The description of visual diagnosis with or without using loupes by student and dentist in Table 2 is contradictory, please correct the mistakes.
2. The authors divided the inspectors into two groups: dentistry student and the dentists (intern and specialist). However, intern and specialist have obvious differences in clinical experience and professional level, so it would be more appropriate to divide the inspectors into three groups.
3. The gold standard was the diagnosis made by the paediatric dentist with loupes according to the article, but the specific results obtained according to the gold standard are not clear in the paper, so it is suggested to supplement the data.
4. The calculation method of the correct diagnosis percentage of occlusal caries in children in Table 3 was improper, because it is necessary to supply correct diagnosis rate based on the gold standard: the number of correct diagnoses based on the gold standard made by students or dentists in Table 3 divided by the total number of examined teeth.
5. The experimental data in Table 4 - 6 should compare the differences between student and dentist in the diagnosis of occlusal caries in children with or without using loupes; the comparison content in experimental design should be focused on the differences between students and doctors instead of those between primary teeth and permanent teeth.
6. To reduce bias, 50% of the tests were performed first without loupes and then with x2.5 loupes without lighting, and the other half were performed first with x2.5 loupes without lighting and then without loupes. However, I think all the test should be done first without loupes and then with x2.5 loupes without lighting, because the first visual examination with x2.5 loupes will affect the judgement of the second visual examination without x2.5 loupes.
In conclusion, I suggest revision.
Author Response
Thanks for the helpful comments. Below is our response.
Introduction must be improved.
The introduction improved. (Attached revised manuscript)
- The description of visual diagnosis with or without using loupes by student and dentist in Table 2 is contradictory, please correct the mistakes.
- corrected
- The authors divided the inspectors into two groups: dentistry student and the dentists (intern and specialist). However, intern and specialist have obvious differences in clinical experience and professional level, so it would be more appropriate to divide the inspectors into three groups.
- Agree. It was practically difficult to make two groups (intern and dentists).At that time the intern had examinations and few of them participated, so we joined them to one group. Mentioned in limitations.
- The calculation method of the correct diagnosis percentage of occlusal caries in children in Table 3 was improper, because it is necessary to supply correct diagnosis rate based on the gold standard: the number of correct diagnoses based on the gold standard made by students or dentists in Table 3 divided by the total number of examined teeth.
-Corrected
4.The experimental data in Table 4 - 6 should compare the differences between student and dentist in the diagnosis of occlusal caries in children with or without using loupes; the comparison content in experimental design should be focused on the differences between students and doctors instead of those between primary teeth and permanent teeth.
Corrected.
5.To reduce bias, 50% of the tests were performed first without loupes and then with x2.5 loupes without lighting, and the other half were performed first with x2.5 loupes without lighting and then without loupes. However, I think all the test should be done first without loupes and then with x2.5 loupes without lighting, because the first visual examination with x2.5 loupes will affect the judgement of the second visual examination without x2.5 loupes
- Your claim is very logical. Our goal was to eliminate the effect of the first test (with and
without loupes) on the second test that could be in done any research.
Reviewer 2 Report
Dear Authors
Thank you for the contribution to the article, but some mayor modifications would be necessary:
ABSTRACT section: There are no conclusions for the second objective.
Introduction section:
- Please update the concept of caries both as a disease and as a lesion. It is NOT an infectious disease.
-Indicate what do you mean by preventing a correct diagnosis, I think the translation to English has not been aproppiated.
Material and Method section: You have not indicated the results of the Kappa Index.
RESULTS section:
In all the tables, please eliminate subscripts a, b, c, and d and indicate in the first row of each table n(%).
DISCUSSION
The first three paragraphs of the discussion are results not discusion. These results need to be discussed and interpreted in more detail.
Thank you very much for your effort.
Author Response
Thank you for your helpful comments. Below is our response.
- ABSTRACT section: There are no conclusions for the second objective
Added (attached revised manuscript)
- Introduction section: Please update the concept of caries both as a disease and as a lesion. It is NOT an infectious disease.
Corrected and updated. First paragraph
- Indicate what do you mean by preventing a correct diagnosis, I think the translation to English has not been appropriated.
I mean earlier diagnosis. Changed
- Material and Method section: You have not indicated the results of the Kappa Index.
Added
- RESULTS section: In all the tables, please eliminate subscripts a, b, c, and d and indicate in the first row of each table n(%).
Corrected
- DISCUSSION: The first three paragraphs of the discussion are results not discussion. These results need to be discussed and interpreted in more detail.
Made a revision of the discussion. Only the first paragraph is the results. The rest is explanation and clinical applications of the results.
Round 2
Reviewer 1 Report
Comments to the Authors
This study aimed to examine the effectiveness of diagnosing occlusal caries in primary and permanent molar teeth in children under the use of loupes by student and dentist. However, there are some mistakes in the article.
The following issues need to be improved and clarified:
1. The group name of visual diagnosis with or without using loupes by student and dentist in Table 2 is reverse, please correct the mistakes. In addition, the Table 2 and Table 3 should be merged into one table. The correct diagnosis rates of occlusal caries in children compared to the gold could be shown in Table 2.
2. The experimental design in Table 4 to Table 6 should compare the difference between student and dentist in the diagnosis of occlusal caries in children with or without using loupes. So, the groups of the statistical calculation were wrongly selected. The difference on diagnosis results between permanent teeth and primary teeth did not reflect the difference on the students' diagnosis with or without using loupes.
3. It is recommended to add statistics results on the overall accuracy of the diagnosis of occlusal caries by student with or without using loupes and by dentist without using loupes. Because,the overall accuracy of the diagnosis of occlusal caries has very import meaning.
4. The results part of the abstract of this paper only describes the results of ICDAS0 and ICDAS1, without mentioning the results of ICDAS2 and ICDAS3, and the results and Clinical significance content did not capture the main idea of the research purpose of this paper. Besides, “The student's examinations without the loupes detected caries” in the abstract should be corrected as “The student's examinations without the loupes detected no caries” and the P value in the sentence of “there was a statistically significant difference (p > 0.047) when the diagnosis was made by a student using loupes” should be p =0.047.
5. The paper was not well organized and written, and the language needs to be modified by a language expert. For example, “Dental caries is a biofilm-mediated, diet modulated, multifactorial, non-communicable, dynamic disease resulting in net mineral loss of dental hard tissues However caries lesion is the clinical sign of caries.” in the introduction.
6. Some paragraphs have two blanks at the beginning, and some didn't. Please unify the format
In conclusion, I suggest major revision.
Author Response
This study aimed to examine the effectiveness of diagnosing occlusal caries in primary and permanent molar teeth in children under the use of loupes by student and dentist. However, there are some mistakes in the article.
The following issues need to be improved and clarified:
- The group name of visual diagnosis with or without using loupes by student and dentist in Table 2 is reverse, please correct the mistakes. In addition, the Table 2 and Table 3 should be merged into one table. The correct diagnosis rates of occlusal caries in children compared to the gold could be shown in Table 2.
Thank you for the comment. We corrected the mistake and merged the tables.
- The experimental design in Table 4 to Table 6 should compare the difference between student and dentist in the diagnosis of occlusal caries in children with or without using loupes. So, the groups of the statistical calculation were wrongly selected. The difference on diagnosis results between permanent teeth and primary teeth did not reflect the difference on the students' diagnosis with or without using loupes.
Thank you for the comment. We now compare students and dentists.
- It is recommended to add statistics results on the overall accuracy of the diagnosis of occlusal caries by student with or without using loupes and by dentist without using loupes. Because,the overall accuracy of the diagnosis of occlusal caries has very import meaning.
Could you please elaborate which index of accuracy do you recommend presenting in the manuscript?
- The results part of the abstract of this paper only describes the results of ICDAS0 and ICDAS1, without mentioning the results of ICDAS2 and ICDAS3, and the results and Clinical significance content did not capture the main idea of the research purpose of this paper. Besides, “The student's examinations without the loupes detected caries” in the abstract should be corrected as “The student's examinations without the loupes detected no caries” and the P value in the sentence of “there was a statistically significant difference (p > 0.047) when the diagnosis was made by a student using loupes” should be p =0.047.
We corrected the relevant text in the abstract. In addition we also elaborated the report in the abstract for ICDAS2 and ICDAS1+2.
- The paper was not well organized and written, and the language needs to be modified by a language expert. For example, “Dental caries is a biofilm-mediated, diet modulated, multifactorial, non-communicable, dynamic disease resulting in net mineral loss of dental hard tissues However caries lesion is the clinical sign of caries.” in the introduction.
We have conducted an additional modification by a language expert.
- Some paragraphs have two blanks at the beginning, and some didn't. Please unify the format
We edited the manuscript with unified blanks.
Reviewer 2 Report
Thank you very much for your effort.
Author Response
Thank you very much for your time and efforts.
Round 3
Reviewer 1 Report
Comments to the Authors
This study aimed to examine the effectiveness of diagnosing occlusal caries in primary and permanent molar teeth in children under the use of loupes by student and dentist. However, there are some mistakes in the article.
The following issues need to be improved and clarified:
1. The group name of correct diagnosis with or without loupes by student in Table 2 was reverse, please correct the mistakes. In addition, the visual diagnosis and correct diagnosis of student or dentist should be clearly marked in Table 2.
2. The data in Table 3 were student and dentist diagnosis without loupes in primary and permanent teeth. However, the title of Table 3 was student and dentist diagnosis using loupes in primary and permanent teeth. This is confusing.
3. In the results section of the abstract, the sentence of “student correctly diagnosed no caries (ICDAS0) in 60% ” is wrong, because the group name of correct diagnosis with or without using loupes by student in Table 2 was reverse. Moreover, the sentence of “The dentist (without loupes) diagnosed 62.5% without the loupes as ICDAS1, and 82.1% as ICDAS0 (p <0.01).” was repeated with the next sentence. Please refine repeated statements.
In conclusion, I suggest revision.
Author Response
Dear Reviewer
Thank you for your fruitful comments.
Below is our response:
- The group name of correct diagnosis with or without loupes by student in Table 2 was reverse, please correct the mistakes. In addition, the visual diagnosis and correct diagnosis of student or dentist should be clearly marked in Table 2
Response first section of note 1
We repeated all the results again and as far as we understood your comment the source of the error is in the text of the results. We should have written “examination without loupes” instead
of macroscopic examination. We changed it in the text
response section 2 of note 1
In addition, the visual diagnosis and correct diagnosis of student or dentist should be clearly marked in Table 2
Corrected: marked in the table 2
- 2. The data in Table 3 were student and dentist diagnosis without loupes in primary and permanent teeth. However, the title of Table 3 was student and dentist diagnosis using loupes in primary and permanent teeth. This is confusing.
Title corrected to
Student and Dentist diagnosis without loupes in primary and permanent teeth.
- 3. In the results section of the abstract, the sentence of “student correctly diagnosed no caries (ICDAS0) in 60% ” is wrong, because the group name of correct diagnosis with or without using loupes by student in Table 2 was reverse. Moreover, the sentence of “The dentist (without loupes) diagnosed 62.5% without the loupes as ICDAS1, and 82.1% as ICDAS0 (p <0.01).” was repeated with the next sentence. Please refine repeated statements.
Moreover, the sentence of “The dentist (without loupes) diagnosed 62.5% without the loupes as ICDAS1, and 82.1% as ICDAS0 (p <0.01).” was repeated with the next sentence. Please refine repeated statements
corrected
Round 4
Reviewer 1 Report
Comments to the Authors
This study aimed to examine the effectiveness of diagnosing occlusal caries in primary and permanent molar teeth in children under the use of loupes by student and dentist. However, there are some mistakes in the article.
The following issues need to be improved and clarified:
The order of the first two groups of data in Table 2 is with loupes and then without loupes. Why the sequence of data in the third group is changed to without loupes followed by with loupes. So, this makes the calculation process of the student correct diagnosis percentage in Table 2 confusing. In addition, the forth group “Dentist correct diagnosis” only has data without loupes, lacking the data with loupes. Severely, the formulas used to calculate the percentages in Table 2-4 are clearly inconsistent and confusing, which were also missing in the methods part. So all the formulas or calculation methods used should be listed in detail. These points have been repeatedly raised in the former review comments, but they still have not been effectively answered.
In conclusion, in view of the author's repeatedly modified results have not effectively responded to the questions raised, we regret that we can only reject it.
Author Response
Dear reviewer:
Thank you for your notes.
Below are our comments:
In table 2 I recommend leaving only the ICDAS score as the title of each column. Also data on the correct diagnosis of dentists using loupes are missing
Corrected
data on the correct diagnosis of dentists using loupes are missing.
The diagnosis of the dentist with loupes was the gold standard, that is 100% correct. Because of that we didn’t write it. We wrote that below the line at the bottom of table 2.